# The Role of Interleukin-17 in Juvenile Idiopathic Arthritis: From Pathogenesis to Treatment

**DOI:** 10.3390/medicina58111552

**Published:** 2022-10-28

**Authors:** Marino Paroli, Luca Spadea, Rosalba Caccavale, Leopoldo Spadea, Maria Pia Paroli, Nicola Nante

**Affiliations:** 1Division of Clinical Immunology, Department of Clinical, Anesthesiologic and Cardiovascular Sciences, Faculty of Medicine, Sapienza University of Rome, 00185 Rome, Italy; 2Post Graduate School of Public Health, University of Siena, 53100 Siena, Italy; 3Eye Clinic, Department of Sense Organs, Sapienza University of Rome, 00185 Rome, Italy; 4Department of Molecular and Developmental Medicine, University of Siena, 53100 Siena, Italy

**Keywords:** interleukin-17, interleukin-23, juvenile idiopathic arthritis, secukinumab

## Abstract

*Background and Objectives*: Interleukin-17 (IL-17) is a cytokine family consisting of six members and five specific receptors. IL-17A was the first member to be identified in 1993. Since then, several studies have elucidated that IL-17 has predominantly pro-inflammatory activity and that its production is involved in both the defense against pathogens and the genesis of autoimmune processes. *Materials and Methods*: In this review, we provide an overview of the role of interleukin-17 in the pathogenesis of juvenile idiopathic arthritis (JIA) and its relationship with IL-23, the so-called IL-23–IL-17 axis, by reporting updated findings from the scientific literature. *Results:* Strong evidence supports the role of interleukin-17A in the pathogenesis of JIA after the deregulated production of this interleukin by both T helper 17 (Th17) cells and cells of innate immunity. The blocking of IL-17A was found to improve the course of JIA, leading to the approval of the use of the human anti-IL17A monoclonal antibody secukinumab in the treatment of the JIA subtypes juvenile psoriatic arthritis (JPsA) and enthesitis-related arthritis (ERA). *Conclusions:* IL-17A plays a central role in the pathogenesis of JIA. Blocking its production with specific biologic drugs enables the effective treatment of this disabling childhood rheumatic disease.

## 1. Introduction

Juvenile idiopathic arthritis (JIA) is a chronic disease with childhood onset that currently has no definitive cure [1]. Thus, the goal of therapy is to control disease activity and prevent complications consequent to the chronic inflammatory state [2,3,4]. In addition to steroid therapy and conventional synthetic disease-modifying antirheumatic drugs (csDMARDs), new therapeutic tools, namely biologics, have become available in the past two decades [5]. These drugs, which target specific pro-inflammatory molecules involved in the pathogenesis of the disease, have made it possible to achieve low disease activity and, more rarely, the remission of the disease in a reasonable percentage of patients. However, there remain numerous unmet needs for the optimal therapeutic management of patients with JIA. Recently, interleukin-17 (IL-17) has been shown to play a previously underestimated role in disease pathogenesis. The purpose of this review is to summarize the recent studies supporting the role of IL-17 in JIA and to highlight how these studies have led to the approval of the use of monoclonal anti-interleukin (IL)-17A antibodies for the treatment of the JIA subtypes enthesitis-related arthritis (ERA) and juvenile psoriatic arthritis (JPsA).

## 2. Molecular Features and Signaling of IL-17

In 1993, a new interleukin was cloned. This interleukin was initially defined as CTLA-8 and later as interleukin-17A (IL-17A) [6,7]. IL-17A showed a sequence homology with an open reading frame of *Herpesvirus saimiri*, a virus with a specific tropism for T cells. Quite surprisingly, molecular analysis revealed that IL-17A had no homology with other known cytokines, being characterized by an unusual cysteine-knot fold structure [8,9,10]. Two years later, the receptor for IL-17A (IL-17RA) was discovered [11,12]. Other cytokines structurally similar to interleukin-17A were then identified. All these molecules have been grouped into a family defined as IL-17, which includes six members from IL-17A to IL-17F. IL-17A and IL-17F can form both homodimers and heterodimers (IL-17A/F), whereas IL-17B, IL-17C, IL-17D, and IL-17E exist only as homodimers [13,14,15,16]. Four other receptors for IL-17 were subsequently identified [17,18]. Thus, the IL-17 receptors known so far represent a total of five members (IL-17A to IL17C), all containing the shared cytoplasmic SEFIR motif (SEF/IL-17R). [9]. This protein engages the multifunctional adapter Act1. Act1 in turn binds to E3 ubiquitin, leading downstream to the recruitment and ubiquitination of TNF-receptor-associated factor 6 (TRAF6). This ultimately triggers nuclear factor κB (NF-κB), the nuclear transcription factors CCAAT-enhancer-binding protein β (C/EBPβ), and mitogen-activated protein kinase (MAPK)-dependent activating protein-1 (AP-1) for the transcription of IL-17 target genes [12,19,20,21,22,23,24,25,26]. IL-17 signaling can act synergistically with other ligands, including cytokines or microbial products, leading to the activation of alternative signaling pathways [27,28,29]. Table 1 shows the different members of the IL-17 family, their respective cellular receptors, and known or supposed transcription factors activated after cytokine/receptor binding.

## 3. The Pro-Inflammatory Function of IL-17

The primary activity of IL-17A, the most studied member of the IL-17 family, is to promote tissue inflammation, contributing to the defense against bacteria, fungi, and parasites [23,30,31]. The IL-17A receptor is expressed in several cell types, including fibroblasts, osteoclasts, osteoblasts, monocytes, and synoviocytes [32,33,34,35,36]. Interleukin-17A, after binding to its receptor, can induce the production of various chemokines, such as CXCL1, 5, 8, 9, and 10 and CCL2 and 20 [26,37,38]. These molecules can in turn attract neutrophils, monocytes, and other pro-inflammatory cells to the inflammatory site [7,39,40,41,42]. IL-17A also stimulates the secretion of pro-inflammatory cytokines such as IL-6, TNF, and IL-1 [7]. Although initial studies suggested the role of this cytokine not only in defense against pathogens but also in the pathogenesis of inflammatory and autoimmune diseases [43,44,45,46], it was only with the discovery of a subpopulation of T lymphocytes capable of producing IL-17A (and therefore termed T helper 17 (Th17) cells) that interest in the pro-inflammatory properties of this cytokine was given a major boost [47]. Th17 cells originate from naive CD4+ T cells in response to molecules in the microenvironment, such as IL-1β, IL-6, and TGFβ [32,48,49,50]. The Th17 cell subset mainly differs from Th1 and Th2 cells in that it produces IL-17A, IL17F, IL-21, IL-22, and GM-CSF [32,51]. Once differentiated, Th17 cells express the receptor for IL-23 (IL-23R), which binds the IL-23 necessary for their survival and proliferation and expression of the retinoic, survival, and proliferation of the retinoic-acid-receptor-related orphan nuclear receptor (ROR) γt-cell-lineage-specific transcription factor [52,53,54,55]. Therefore, the term “IL-23–IL-17 axis” was coined to emphasize the dependence of IL-17 production on induction by IL-23 [56,57,58]. More recently, it has been shown that IL-17 can also be produced by other cell types. These include CD8+ cytotoxic T cells (Tc17) [59]; cells belonging to both the adaptive and innate immune system, such as γδ T cells; and cells entirely belonging to the innate system, such as innate lymphoid cells-3 (ILC3), invariant natural killer T (iNKT) cells, NK cells, neutrophils, and macrophages [60,61,62,63,64,65,66,67]. Although the main function of IL-17A is host defense against pathogens, if the production of this cytokine occurs in an excessive and dysregulated manner and affects improper targets, its activity can result in pathological inflammation and autoimmunity. Figure 1 shows the main soluble factors and cell types involved in the pro-inflammatory response of the IL-23–IL-17 axis. The first demonstration of the role of IL-17 in human pathology was provided by the observation that monoclonal antibodies specific to IL-17A were highly effective in the therapy of psoriasis, a skin pathological condition characterized by dysregulated IL-17A production [68]. Following success in the treatment of psoriasis, IL17 blockade was tested in the treatment of inflammatory rheumatic diseases in which members of the IL-17 family had been shown to have a pathogenic role. After an initial failure in the treatment of rheumatoid arthritis by blocking the activity of IL-17A/IL-17F [69,70], anti-IL-17 biologics have proven highly effective in the treatment of psoriatic arthritis (PsA) [71,72] and ankylosing spondylitis (AS) [73,74] in adults. It must be underlined, however, that there are some plausible reasons for the limited success of IL-17 blockade in rheumatoid arthritis. Designing trials to evaluate the effect of an IL-17A blockade in specific groups of RA patients, e.g., those with high levels of IL-17A or Th17 cells in synovial fluid/tissues, those with rapidly progressing erosive disease, or those with early RA and elevated CRP, could yield different results [75]. The importance of an early and effective therapeutic intervention in PsA and AS is evidenced by the observation that if these diseases are left untreated they can lead to severe joint complications such as total hip replacement and subsequent long periods of rehabilitation [76,77]. Interestingly, in AS, the blockade of IL-23 was not as effective as that of IL-17A [74], demonstrating that in such rheumatic diseases IL-17A can also be produced by IL-23-independent cells. In recent years, it has emerged that members of the interleukin-17 family are crucially involved in the pathogenesis of some subtypes of juvenile idiopathic arthritis (JIA). It should be noted that genetic studies have found a significant relationship between other members of IL-17 and JIA. In this regard, a significant relationship was recently demonstrated between the 7488A/G and 7383A/G polymorphisms of IL-17F, resulting in an increased risk of developing severe forms of the disease [78]. In the following sections of this review, we will discuss the role of IL-17 in the pathogenesis of JIA and the therapeutic possibilities provided by blocking members of this cytokine family.

## 4. Classification of JIA Subtypes

JIA is defined as an arthritis of unknown etiology that affects children before the age of 16 and persists for at least six weeks after the exclusion of any other possible causes of joint pathology [1]. The condition differs from other forms of arthritis in children according to several clinical features and causes significant morbidity and disability in pediatric patients [79,80,81,82,83,84]. The pathogenesis of JIA has not yet been fully elucidated [85]. JIA has been classified into several categories, each of which has a specific group of clinical manifestations, genetic background, and etiopathogenesis. Specifically, the International League of Associations for Rheumatology (ILAR) identifies seven main subtypes based on clinical and laboratory criteria presented in the first six months of the disease [79], although new classification criteria have recently been proposed [86]. The classic subtypes include: (1) systemic (sJIA); (2) oligoarthritis; (3) rheumatoid factor (RF)-negative polyarthritis; (4) RF-positive polyarthritis; (5) juvenile psoriatic arthritis (JPsA); (6) enthesitis-related arthritis (ERA); and (7) undifferentiated arthritis (UA). 

## 5. The Current Treatment of JIA

Treatment is aimed at achieving disease remission or at least low disease activity. The standard treatment consists of both physical and drug therapy. The drugs commonly used are intra-articular or systemic steroids [87]; non-steroidal anti-inflammatory drugs (NSAIDs); and immunosuppressive drugs such as methotrexate, leflunomide, and salazopyrin [88,89]. In recent years, due to increased knowledge about JIA, new drugs such as monoclonal antibodies or soluble receptors obtained by engineering living cell cultures and therefore defined as biological have been introduced in therapy. These drugs previously approved for the treatment of adult inflammatory rheumatic diseases can specifically block different factors involved in the pathogenesis of the disease. The main targets of biologics are tumor necrosis factor-alpha; interleukin-1; interleukin-6; T lymphocyte co-stimulatory molecule CTLA-4; and, more recently, interleukin-17A. PD1/PD-L1 molecules are recently discovered biologic drugs whose blockade has fundamentally changed cancer immunotherapy [90]. In Table 2, the currently approved biologics for the different JIA subtypes are shown. 

## 6. The Role of IL-17 in Oligoarticular, Polyarticular, and Enthesitis-Related Arthritis Subtypes

The first evidence that interleukin-17 could play a key role in the pathogenesis of JIA emerged from research in 2007. The authors, using multiplex immunoassay technology, reported in a small sample of patients with JIA that those with the oligoarticular and polyarticular forms had elevated plasma levels of this cytokine in the active phase of the disease [91]. These data were confirmed in a recent study in patients with both the oligoarticular and rheumatoid factor (RF)-negative polyarticular form [92], leading to speculation that these two subtypes of JIA may represent a continuum of the same disease [93]. Next, the first detailed analysis of T cells capable of producing IL-17 in patients with JIA was provided by [94]. In that study, IL17+ T cells were found to be enriched within the joints of the patients when compared with the patients’ peripheral blood and control subjects. This subset of CD4+ T cells exhibited a memory-cell phenotype and was present particularly in the extended oligoarticular form of JIA but not in the persistent form of the disease. An inverse relationship between IL-17+ T cells and FOXP3+ T cells exerting regulatory activity (Treg) was also described in the same study, suggesting a possible explanation for the persistence of arthritic damage due to the reduction of cells with anti-inflammatory activity. This result is particularly relevant given the role of regulatory cells, as previously described in the suppression of joint inflammation in patients with JIA [95]. In another study published in the same year, it was shown that IL-17 levels were increased in the synovial fluid of patients with enthesitis-related arthritis (ERA) and in the polyarticular form as compared to subjects with sJIA [96]. IL-17 levels were directly correlated with disease activity and were not correlated with those of other soluble pro-inflammatory molecules. The high local concentration of IL-17 suggested that cells capable of producing this cytokine were present in significant numbers within the synovial membrane. A predominance of the Th17 lymphocyte population was described in a study conducted by both flow cytometry and enzyme immunoassay in the synovial fluid of children with ERA [97]. The relationship between Th17 and JIA was then further investigated. It was shown in another study that the expression of RORC2, a transcriptional factor of Th17, was significantly increased in the synovial fluid of patients with JIA, and its expression was inversely correlated with that of FOXP3+ mRNA. These results emphasized the possible inverse relationship between Th17 and Treg in JIA patients. However, it was not possible to clarify whether this inverse relationship could be caused by at least the partial exhaustion of Th17 cells in the synovial fluid [98]. The phenotype and function of CD4+ cells present in the synovial fluid of patients with an oligo-articular form of JIA were further investigated. It was therefore reported that the number of CD4+ CD161+ cells with a Th17 signature was higher in the synovial fluid of patients with JIA in the active phase in comparison to the inactive state of the disease. Such cells obtained from synovial fluid but not from peripheral blood were able to change their phenotype from Th17 to Th1 or Th17/Th1 in vitro. This study underlined, therefore, the role of Th17 cells in the pathogenesis of JIA [99,100]. In one study, Th17 cells were found not only in synovial fluid but also in the peripheral blood of patients with active JIA [101]. It was shown in another study that in patients with JIA, the production of this cytokine may be independent of T cell receptor (TCR) stimulation in intra-articular αβT cells with either a CD8+ or CD4/CD8 double-negative phenotype and expressing the surface molecule CD31. IL-17A production occurred as a result of the binding of CD31 by the CD38 molecule [92]. It has also been shown that natural cytotoxicity receptor (NCR)+ or NCR- ILC3 cells are expanded among the mononuclear cells present in the synovial liquid of patients with JIA. The increase in the number of NCR-ILCRs was associated with an increase in CD4+, CD8+, and γδT cells [102]. These results emphasize how different cell types in addition to Th17 cells participate in IL-17 production in patients with JIA. 

## 7. The Potential Role of IL-17 in Systemic JIA

Attention has also been paid to systemic JIA (sJIA). This particular form of JIA with possible autoinflammatory pathogenesis is considered the counterpart of adult Still’s disease (AOSD) [85]. In a small study, increased Th17 cells were reported in the peripheral blood of patients with sJIA [103]. In a later study, similar results were reported, showing the increased expression of IL-17A in circulating γδT cells [104]. It has recently been reported that acute sJIA is characterized by the expansion of IL-17-expressing Treg cells showing a prominent genetic signature of Th17 cells. These cells are likely to lose their suppressive function on inflammation. Interestingly, the occurrence of this genetic signature was dependent on interleukin-1 activity. Due to their plasticity, Th-17 cells can be reprogrammed in an appropriate cytokine microenvironment to generate effector-type T cells (Teff) in patients with chronic diseases [105]. These results suggest that there may be a “window of opportunity” during which IL-1 blockade can inhibit inflammation and progression to chronicity in patients with sJIA.

## 8. The IL-23–IL-17 Axis: The Blockade of IL-23 in the Therapy of JIA

Based on the studies that showed the role of IL-17 in the pathogenesis of JIA and, in particular, the role of the “IL-23–IL-17” axis, clinical studies were initially conducted to evaluate the effect of anti-IL23 biologics in the treatment of this disease. In particular, ustekinumab, a human monoclonal antibody that selectively blocks the common p40 subunit of IL-12 and IL-23, preventing their binding to their membrane receptor [106], has been considered for therapy. Ustekinumab has been shown to be effective in several adult inflammatory diseases, including psoriasis, psoriatic arthritis, and Crohn’s disease [107,108,109]. In a retrospective single-center study analyzing data from patients with the JIA subtype ERA in whom both conventional therapy with DMARDs and two subsequent treatments with anti-TNF-alpha biologics had failed, the global assessment of disease activity by a physician decreased in four of the five patients treated with ustekinumab. A reduction in the number of joints with active inflammation and the improvement of enteritis were observed in three and two patients, respectively. The resolution of sacroiliitis was observed in three patients [110]. Ustekinumab has recently been approved by the FDA for the treatment of children with active psoriatic arthritis. 

## 9. The Role of Anti-IL-17A Blocking Antibodies in JPsA and ERA Subtypes of JIA

Regarding the use of anti-IL-17 biologics, the investigations have focused on secukinumab, a human anti-IL-17A antibody that directly blocks the binding of this cytokine to its receptor [111]. Secukinumab is effective in several inflammatory rheumatic diseases in adults. In particular, the use of secukinumab has been approved for the treatment of psoriatic arthritis and ankylosing spondylitis, including non-radiographic axial spondylitis [73,112,113]. Psoriatic arthritis is a form of juvenile idiopathic arthritis (JIA) and is characterized by chronic joint inflammation and swelling, as well as an increased risk of the asymptomatic inflammation of the eyes [114]. The enthesitis-related arthritis (ERA) category of JIA describes a heterogeneous group of children, including those with enthesitis, arthritis, and inflammatory bowel disease (IBD)-associated arthropathy. ERA accounts for about 15–20% of JIA cases and has a peak age of onset of 12 years [115]. PsA and adult non-radiographic axial spondylitis are considered the respective counterparts of the JIA subtypes juvenile psoriatic arthritis (JPsA) and enthesitis-related arthritis (ERA) [86,116]. Given the pathogenic role of interleukin-17 and, in particular, member IL-17A in these forms of JIA (as demonstrated in the studies described above), as well as the limited effectiveness of currently available therapies (including JPsA [117] and ERA [118,119,120,121,122]), it was hypothesized that secukinumab could be successfully used in the treatment of these two forms of JIA. In a retrospective study that analyzed patients who had already been treated using biologics with other mechanisms of action, it was found that patients with JIA showed significant improvements in signs and symptoms in different disease domains [123]. In a recent phase-three study [124], the administration of this biologic was shown to reduce the frequency of disease flare-ups in children with JIA compared with control subjects. In more detail, the 2-year, randomized-withdrawal, double-blind, placebo-controlled JUNIPERA trial included 86 patients and consisted of three treatment periods. In treatment period 1 (TP1), all eligible subjects entered the trial to receive 12 weeks of open-label secukinumab at a dose predicted to achieve secukinumab serum levels equivalent to those in adults administered with a 150 mg dose regimen. Secukinumab was administered subcutaneously weekly for the first 4 weeks and every 4 weeks thereafter. Clinical response (JIA ACR 30) was assessed at Week 12. Responders advanced to TP2 and non-responders exited the trial and entered into the post-treatment follow-up period. In TP2, subjects who were responders (defined as achieving JIA ACR 30 at Week 12) entered the double-blind withdrawal period and were randomly assigned 1:1 to either secukinumab or placebo on that visit and then every 4 weeks, until they either experienced a disease flare-up or completed TP2. TP2 was event-driven and was planned to be closed when 33 subjects experienced a disease flare-up, as per the JIA definition. Alternatively, the study could be closed when all subjects reached the total study duration of 104 weeks, and therefore the subjects who did not experience a disease flare-up remained in TP2 for the duration of the study and completed the study without entering into TP3. Subjects experiencing a disease flare-up in TP2 immediately entered TP3 to receive open-label secukinumab every 4 weeks until the total study duration of 104 weeks for that subject was achieved. A post-treatment follow-up period lasting 12 weeks from the final study drug administration was required for all subjects, unless they qualified and entered the secukinumab extension trial. Patients were initially included if they presented with acute-phase JPsA or ERA, had not been treated previously with biologics, and showed inadequate response to standard therapy. The dosage used ranged from 75 to 150 mg monthly after an induction period with the weekly administration of secukinumab for 4 weeks. The study results showed the rapid improvement of several clinical domains such as arthritis, dactylitis, and enthesitis in treated children compared with the control group. Importantly, the risk of flare-ups was found to be decreased by 72%. In addition, the goal of achieving inactive disease was achieved in 40% of subjects throughout the study. These studies support the conclusion that blocking IL-17A activity by secukinumab is a safe and effective treatment in patients with the JPsA and ERA subtypes of JIA. Based on the results of the JUNIPER study, secukinumab was recently approved by both the FDA and EMA for the treatment of JPsA and ERA in children aged ≥6 years. In Table 3, the currently available biologics that selectively block the IL-23–IL-17 axis are shown.

## 10. Conclusions

This review emphasized the key role of the interleukin-17 cytokine family and, in particular, IL-17A in the pathogenesis of JIA. Treatment with IL-17A-blocking antibodies such as secukinumab has provided promising results in the treatment of children with JPsA and ERA [78]. Studies on the pathogenesis of the disease have also emphasized the possible role of IL-17 in other forms of JIA, including sJIA. Therefore, IL-17 blockade may be extended in the future to other subtypes of JIA. The availability of therapies based on new molecules capable of blocking other members of the IL-17 family or it could further expand the therapeutic opportunities. Multicenter, randomized, double-blind studies and real-world studies including a large number of patients are needed to clarify the role of anti-IL-17 therapy in the treatment of this severely disabling pediatric disease.

## Figures and Tables

**Figure 1 medicina-58-01552-f001:**
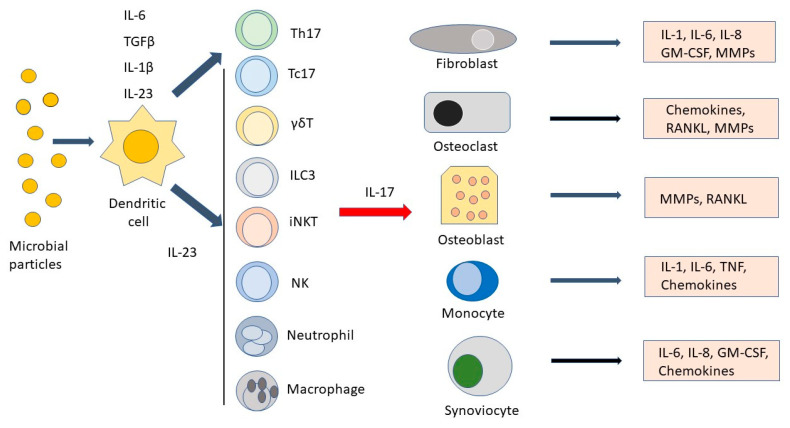
Main cell types and soluble factors involved along the interleukin-23 (IL-23)/IL17 axis. Microbial products are phagocytosed by dendritic cells, which, in turn, induce T helper 17 (Th17) cells to differentiate in the presence of interleukin-1β (IL-1β), IL-6, and tissue-growth factor-β (TGFβ). IL-23 ensures Th17 cell survival and stimulates IL-17 production by Th17 and other cell types such as T cytotoxic 17 (Tc17) cells, γδT cells, innate lymphoid cells type 3 (ILC3), invariant natural killer cells (iNKT), NK cells, neutrophils, and macrophages. After stimulation by IL-17, fibroblasts, osteoclasts, osteoblasts, monocytes, and synoviocytes produce IL-1, IL-6, IL-8, granulocyte macrophage colony-stimulating factor (GM-CSF), chemokines, tumor necrosis factor (TNF), and matrix metalloproteinases (MMPs), leading to the amplification of the inflammatory response.

**Table 1 medicina-58-01552-t001:** The interleukin-17 family members, receptors, and target transcription factors.

Cytokine	Receptor	Activated Transcription Factor
IL-17 A/A	IL-17RA/IL-17RC IL-17RC/IL-17RC IL-17RA/IL-17RD	C/EPBβ, AP-1, NF-κB (NF-κB) (NF-κB)
IL-17 F/F	IL-17RA/IL-17RC IL-17RC/IL-17RC	NF-κB (NF-κB)
IL-17A/F	IL-17RA/IL-17RC IL-17RC/IL-17RC	NF-κB (NF-κB)
IL-17B/B	IL-17RA/IL-17RB	C/EPBβ, AP-1, NF-κB
IL-17E/E (IL-25)	IL-17RA/IL-17RB	C/EPBβ, AP-1, NF-κB
IL-17C/C	IL-17RA/IL-17RE	NF-κBζ
IL-17D	CD93	Intra-cellular CD93 domain

C/EPBβ = CCAAT-enhancer-binding protein β; AP-1 = activator protein-1; NF-κB(ζ) = nuclear factor kappa-light-chain-enhancer of activated B cells (ζ). Transcription factors in parentheses have not yet been conclusively demonstrated.

**Table 2 medicina-58-01552-t002:** Biologics currently approved for JIA therapy.

Biologic	Mechanism of Action	JIA Subtype	Approving Agency
Etanercept	Binding to TNFα	pJIA, JPsA, ERA	FDA/EMA
Adalimumab	Binding to TNFα	pJIA, ERA	FDA/EMA
Golimumab	Binding to TNFα	pJIA	FDA/EMA
Tocilizumab	Binding to IL6R	sJIA, pJIA	FDA/EMA
Anakinra	Binding to IL-1Ra	sJIA	EMA
Canakinumab	Binding to IL-1β	sJIA	FDA/EMA
Abatacept	Binding to CD80/CD86	pJIA	FDA/EMA
Ustekinumab	Binding to IL-23/IL-12	JPsA	FDA
Secukinumab	Binding to IL-17A	JPsA, ERA	FDA/EMA

pJIA = polyarticular juvenile idiopathic arthritis (JIA); JPsA = juvenile psoriatic arthritis; ERA = enthesitis-related arthritis; sJIA = systemic JIA.

**Table 3 medicina-58-01552-t003:** Approved anti-IL23 and anti-IL-17 biologics for adult indications.

Biologic	Mechanism of Action	Approved Indication
Ustekinumab	Binding to IL-23/IL-12	PsO, PsA, CD, UC
Guselkumab	Binding to IL-23	PsO, PsA
Tildrakizumab	Binding to IL-23	PsO
Risankizumab	Binding to IL-23	PsO, PsA, CD
Secukinumab	Binding to IL-17A	PsO, PsA, AS, nr-axSpA
Ixekizumab	Binding to IL-17A	PsO, PsA
Brodalumab	Binding to IL-17RA	PsO

PsO = psoriasis; PsA = psoriatic arthritis; CD = Crohn’s disease; UC = ulcerative colitis; AS = ankylosing spondylitis; nr-axSpA = non-radiographic axial spondiloarthritis.

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
