# Peer review of "The Role of Interleukin-17 in Juvenile Idiopathic Arthritis: From Pathogenesis to Treatment"

_medicina, 2022, doi:10.3390/medicina58111552_

Round 1
Reviewer 1 Report
Comments can be found in the attached pdf file

Author Response
Reviewer#1:
- Query: Table1: I would include the references to where you got each piece of information in this table. Answer: Table 1 has been moved to the second section of the manuscript. The text has been checked and missing references have been added.
- Query: Interleukin-17A after binding to its receptor can induce the production of various chemokines such as CXCL1, 5, 8, 9, and 10, CCL2, and 20. Please add reference. Answer: Relative reference has been added
- Query: rheumatoid arthritis. There are however some studies which have shown that IL-17 blockade in RA does work, but that these patients weren't sufficiently stratified. Answer: “rheumatoid” has been corrected to “rheumatoid arthritis”. Regarding your important point, it has been included in the text and relative reference has been added.
- Query: “that often require”. Please change to “such as”. Answer: The suggested change has been made
- Query: “genesis”. Change to “pathogenesis”: Answer: The suggested change has been made.
- Query: The role of interleukin-17 in JIA. Although the content in this section is fine, please consider re-structuring into a number of different paragraphs - it is too long. Answer: The text has been cut into smaller parts to make it more understandable.
- Query: Inactivity? Better “low disease activity”. Answer: We agree with your observation. The term "inactivity" has been changed to "low disease activity"
- Query: Such Th17 signature (doesn't quite make sense, please re-word). Answer: The sentence has been rephrased to be understandable.
- Query: 4. Blockade of IL-17 activity in the treatment of JIA: This section also needs to be split into a number of paragraphs to make it easier to follow. Answer: This section has been divided into smaller parts to make it more readable.
Reviewer 2 Report
Reviewer comments
Title The role of interleukin-17 in juvenile idiopathic arthritis: from pathogenesis to treatment
The topic is interesting and well written .However some points need to be addressed
|
Position |
Manuscript |
Comments |
|
Paragraph 1 introduction |
Table 1 shows the different members of the IL-17 family |
It is not usual to add a table in the introduction which represent The introduction is usually focus on the problem, and why should anyone care |
|
Introduction |
|
The authors should refer to the research gap before the aim of the review |
|
The pro-inflammatory function of IL-17 |
|
the simplest writing style is usually best These too long paragraphs should be divided to be smaller and informative It will be better if the authors add an illustrative figure on this point
|
|
3. The role of interleukin-17 in JIA |
In that study IL17+, T cells |
Revision of grammar and English editing are warranted Again ,too long paragraphs |
|
I think adding such references to the manuscript may be valuable Association between Interleukin-17F 7488A/G and 7383A/G polymorphisms and susceptibility to juvenile idiopathic arthritis
|
||
Author Response
Reviewer#2:
- Query: It is not usual to add a table in the introduction section. Answer: The table has been moved from the introduction to the next section of the manuscript.
2) Query: The introduction is usually focused on the problem, and why should anyone care. Answer: The introduction has been extended and modified accordingly.
3) Query: The authors should refer to the research gap before explaining the aim of the review. Answer: The gap in research on the role of IL-17 in JIA was highlighted in the introduction section.
- Query: The simplest writing style is usually best. These too long paragraphs should be divided to be smaller and informative. Answer: The paragraphs have been divided into smaller parts to be better readable and understandable. Additional information has been added when needed for better information on the topic.
- Query: The pro-inflammatory function of IL-17: it would be better if the authors might add an illustrative figure on this point. Answer: A figure (figure 1) has been added illustrating the IL-17-patway and its pro-inflammatory activity on target tissues/organs.
- Query: Revision of grammar and English editing are warranted. Answer: English has been edited extensively and grammar errors corrected.
- Query: Again, too long paragraphs. Answer: The paragraphs have been divided into smaller parts throughout the manuscript.
- Query: I think adding such references to the manuscript may be valuable: “Association between Interleukin-17F 7488A/G and 7383A/G polymorphisms and susceptibility to juvenile idiopathic arthritis”. Answer: This reference has been added to the manuscript and commented on.
Reviewer 3 Report
The manuscript is interesting in regard to the topic addressed.
The metodological approach is correct and the conclusions are in line with the results obtained.
It could be worthy of publication, being very relevant to the important issue, It is advisable to mention PMID: 33261292 after talking about "The main targets of biologics are tumor necrosis factor-alpha, interleukin-1, interleukin-6, T lymphocyte co-stimulatory molecule CTLA-4, and more recently interleukin-17A"
Author Response
Reviewer#3:
1) Query: It is advisable to mention PMID: 33261292 after talking about "The main targets of biologics are tumor necrosis factor-alpha, interleukin-1, interleukin-6, T lymphocyte co-stimulatory molecule CTLA-4, and more recently interleukin-17A". Answer: The reference has been added where suggested by the reviewer.
Round 2
Reviewer 2 Report
The authors addressed all the required modifications
Author Response
Thank you for appreciation of our manuscript